# Climate Change Impacts on the Potential Distribution of *Apocheima cinerarius* (Erschoff) (Lepidoptera: Geometridae)

**DOI:** 10.3390/insects13010059

**Published:** 2022-01-05

**Authors:** Weicheng Ding, Hongyu Li, Junbao Wen

**Affiliations:** 1College of Forestry, Beijing Forestry University, Beijing 100083, China; weichengding@bjfu.edu.cn (W.D.); hongyuli@bjfu.edu.cn (H.L.); 2Beijing Key Laboratory for Forest Pest Control, Beijing Forestry University, Beijing 100083, China

**Keywords:** *Apocheima cinerarius* (Erschoff), bioclimatic model, potential distribution, CLIMEX, niche model

## Abstract

**Simple Summary:**

*Apocheima cinerarius* (Erschoff) is an important forest pest in China. It has many hosts and occurs in 20 provinces of China, causing huge economic and ecological losses. The northern temperate zone (north latitude 30–50° N) and the south temperate zone (south latitude 20–60° S) are areas suitable for this pest, although with irrigation, suitability is reduced and the suitable area is smaller, and so superimposed irrigation will make the suitable area smaller. With projected climate change, the suitable area of *A. cinerarius* will move northward. A model based on climate and available hosts for *A. cinerarius* indicates potential for its diffusion and colonization. This research provides a theoretical basis for preventing and controlling the invasion and spread of *A. cinerarius*.

**Abstract:**

Among the impacts of ongoing and projected climate change are shifts in the distribution and severity of insect pests. Projecting those impacts is necessary to ensure effective pest management in the future. *Apocheima cinerarius* (Erschoff) (Lepidoptera: Geometridae) is an important polyphagous forest pest in China where causes huge economic and ecological losses in 20 provinces. Under historical climatic conditions, the suitable areas for *A. cinerarius* in China are mainly in the northern temperate zone (30–50° N) and the southern temperate zone (20–60° S). Using the CLIMEX model, the potential distribution of the pest in China and globally, both historically and under climate change, were estimated. Suitable habitats for *A. cinerarius* occur in parts of all continents. With climate change, its potential distribution extends northward in China and generally elsewhere in the northern hemisphere, although effects vary depending on latitude. In other areas of the world, some habitats become less suitable for the species. Based on the simulated growth index in CLIMEX, the onset of *A. cinerarius* would be earlier under climate change in some of its potential range, including Spain and Korea. Measures should anticipate the need for prevention and control of *A. cinerarius* in its potential extended range in China and globally.

## 1. Introduction

Poplar looper (*Apocheima cinerarius* (Erschoff) (Lepidoptera: Geometridae)) is a polyphagous forest pest in northern China [1]. *Apocheima cinerarius* was originally endemic to Xinjiang, but because of the impacts of quaternary climatic variations, *A. cinerarius* moved eastward and became dispersed throughout the whole of northern China [2]. Phylogenetic trees with complete mitochondrial genomes indicated that there were two lineages of *A. cinerarius* one in the east and one in the west. The two lineages overlap in Gansu Province [3]. Larvae of *A. cinerarius* have a host range of over 20 plant species, mainly harming *Populus*, *Ulmus*, *Salix*, *Sophora japonica* (Linn.), *Elaeagnus angustifolia* (Linn.), *Malus pumila* (Mill.), and sometimes even some of the annual crops in adjoining fields [4]. Females of *A. cinerarius* are not capable of flight and the species is univoltine. It overwinters as a pupa in the soil and emerges from early March to early May, crawling to a host tree to mate and oviposit. The hatched larvae climb up the branches to feed on the leaves. In northern China, *A. cinerarius* can consume entire tree canopies in a matter of days when it is severe, wreaking havoc on the forest landscape and ecosystem [5]. Plantations have been constantly threatened by *A. cinerarius* in northern China [6].

Euphrates poplar (*Populus euphratica* (Oilv.)) is a medium-sized deciduous tree that grows in dry and semi-arid locations. It is an essential constructive tree species in desert environments because of its unique physiological properties that make it tolerant to salty and brackish water and resistant to droughts and sandstorms. It serves as a crucial barrier in protecting oases and maintaining the stability of the oasis ecosystem [7,8]. In 2006, the area of *P. euphratica* forest affected by *A. cinerarius* reached 1.4 million hectares, accounting for about 31.56% of the area of the poplar forest in Xinjiang. In the Tarim Basin, 73.6% of the forest area was impacted. Attacks by *A. cinerarius* weaken the tree. Severe attacks by this pest can kill the tree and if this occurs over large areas this can lead to reduced biodiversity [6].

The adult emergence rate of *A. cinerarius* is closely related to the air temperature, ground temperature, and relative soil humidity [9]. As poikilotherms, insects are sensitive to increases in temperature caused by climate change [10]. Climate change will extend the geographic range of many insect species, and accelerate insect growth and development, resulting in earlier seasonal occurrence. This can change the synchronization between insects and host plants [11]. Temperature interacts with other elements including rainfall, humidity, irradiance, and carbon dioxide concentrations [12]. These changes may make previously unsuitable northern areas suitable habitat for some pests. For example, the predicted rise in temperature will result in more degree days accumulating over shorter periods, and the temperature will surpass the estimated lower threshold for *Thrips tabaci* growth, this will accelerate the growth of *Thrips tabaci* and make it more difficult to predict the impact of the damage it will cause [13]. As another example, warmer temperatures will help *Dendrolimus superans* to develop faster and survive longer than normal, resulting in more insects surviving the winter and facilitating the outbreak the following year [14]. Therefore, it is of great significance to clarify the impact of climate warming on the potential suitable distribution of *A. cinerarius*.

Anthropogenic trade can shift species beyond their native range and break down biogeographic barriers [15]. Increased drought duration and altered rainfall patterns due to climate change have significant effects on species reproduction, survival, dispersal, and population dynamics, which ultimately determine the potential for invaders to move into new communities [16,17]. Determining the geographical distribution of pests under climate change scenarios is critical to formulating long-term management strategies [18]. Potential geographic distribution quantitative prediction models and software mainly include location comparison models CLIMEX, BIOCLIM, and DIVA-GIS, the MaxEnt model and software and the GARP model and software [19,20]. The CLIMEX model outperforms correlative models because it takes into account physiological rather than just occurrence data from the species and enables seasonal phenology modeling [21].

Some scholars have used the MAXENT model to predict the suitable area for *A. cinerarius* in Xinjiang, China [22], but they have not yet conducted research on the suitable area on a global scale. The influence of future climate change on its possible geographic dispersion is critical for scientifically based prevention and management. Based on current and future climatic data, as well as known distribution and biological data, this study employs the CLIMEX model to anticipate the probable geographic distribution of the insect on a worldwide scale in order to give a degree of quarantine, control, and theoretical foundation.

## 2. Materials

### 2.1. CLIMEX Model

CLIMEX 4.0.2 is dynamic simulation software that may be used to forecast plant, animal, and disease distributions. We constructed an ecoclimatic index (EI) ranging from 1 to 100 using “Compare the Location” to represent the climate appropriateness of a certain species at a given location [23]. The analytical procedure is similar to niche modeling, which objectively represents organism interactions in the environment [24]. The growth index (GI), stress index (SI), and restricted circumstances are all elements that influence the value of EI (diapause and minimum heat accumulation during the growing season). The annual growth index (GIA) is used to define the population growth potential under different growth circumstances, which is mostly influenced by the temperature index (TI) and the moisture index (MI) [25]. The stress index, which represents the surviving extremes of temperature and humidity, allows the distribution to be determined in terms of bad seasonal circumstances. The EI value is determined not only by the growth and stress indices but also by effective the accumulated temperature (PDD), the diapause index (DI), and other aspects [26].

### 2.2. Known Distribution Data

Due to a lack of relevant species records in multiple databases, general data regarding the present global distribution of *A. cinerarius* were collected from the literature. This pest is mostly distributed in Beijing, Tianjin, Heilongjiang, Liaoning, Inner Mongolia, Shandong, Shanxi, Hebei, Shaanxi, Ningxia, Gansu, Xinjiang, Tibet, and other provinces [27]. There are also records in Central Asia, such as Uzbekistan, Kazakhstan, and Kyrgyzstan [28]. We removed records with missing geographical coordinates and duplicates from GBIF. The published literature was used to determine the known occurrences. We geo-referenced the points based on the place names supplied in articles containing location information but no coordinate data. Points with locations recorded in the literature but without latitude and longitude coordinates are determined by Google Earth.

### 2.3. Meteorological Data and Irrigation

Meteorological data for running the CLIMEX model were obtained from CliMond (10′ gridded dataset). The current climate fit with baseline climate data (the daily minimum and maximum temperature (°C), the relative humidity (%), monthly precipitation (Mm) at 09:00 and 15:00 h (local time)) centered on 1975. Simulations of future climate uses the “CSRIO A1B Group” datasets. CliMond provides future climate data for different periods (2030, 2050, 2070, 2080, 2090, and 2100), and the composition of the data set is consistent with the current climate data. In the future dataset, 2030 represents the average of the years 2030–2039 (which can also be represented by 2035), and other years are similar [29,30].

*Apocheima cinerarius* oversummers and overwinters in the pupal period and does not emerge until the next spring [31]. Irrigation changes soil moisture and affects the emergence rate of pupae, thereby limiting the distribution of species to varying degrees. We obtained global irrigation data from the Food and Agriculture Organization of the United Nations (https://www.fao.org/faostat/en/#home) (accessed on 5 November 2021) According to the global irrigation area reported by Siebert [32], we divided the irrigation scenarios into two conditions. The first is to use a supplementary irrigation scheme of 1.5 mm/day in summer, and the second is to combine the EI value of the irrigated area with the natural rainfall area. The EI value was superimposed to obtain the final prediction map of the suitable area in the global irrigation area.

### 2.4. CLIMEX Parameter

The main distribution area of *A. cinerarius* in China has a temperate continental climate. A semi-arid and obligate diapause template that comes with the CLIMEX software combines biological data and geographical distribution adjustment parameters. The final parameter values are shown in Table 1, and how they were obtained is explained in the following sections.

#### 2.4.1. Temperature Index

The parameters were set according to experimental results and geographical records. Based on Wang [33], the lower temperature threshold (DV0) was 0 °C, but Qing [34] found that due to the unusual weather in 2015, *A. cinerarius* will continue to grow at −2 °C. In order to better match the adaptability of regions with lower temperatures, we set DV0 to −2. Wu [28] showed that the temperature range during the peak period emergence was 6.5–14.5 °C, so DV1 and DV2 were set to 7 °C and 15 °C, respectively. The occurrence period is mainly concentrated in February, March, and April. Based on the work of Liu [35] and combined with the temperate template (Built-in software), we set DV3 to 25 °C.

#### 2.4.2. Moisture Index

*Populus euphratica*, an important host of *A. cinerarius*, can tolerate drought and salinity. Referring to the drought template, we set SM0 to 0.05. The emergence rate is high when the soil moisture is 10–30%. As the humidity increases, the emergence rate shows a downward trend, and the decline is greatest between 40–50%; thus, we set the SM1, SM2, and SM3 to 0.1, 0.4, and 0.45, respectively [36].

#### 2.4.3. Stresses

*Apocheima cinerarius* suspends its growth after being affected by low temperature [37]. Combined with the template, we set the TTCS to −5 and the THCS to 0.0002. According to DV3, TTHS was set to 25 °C and combined with the temperate template, while THHS was set to 0.005. SMDS and HDS were set to 0.05 and 0.005, respectively, according to SM0 and the template. According to SM3 and template data, SMWS and HWS were set to 0.45 and 0.002, respectively.

#### 2.4.4. Effective Accumulated Temperature

The effective accumulated temperatures of eggs, larvae, pupae, and adults are 235.4 ± 26.8, 368.46 ± 29.6, 55.62 ± 99.4, and 56.06 ± 14.8, respectively. We set the effective accumulated temperature to 650 [38].

#### 2.4.5. Diapause

*Apocheima cinerarius* diapause in summer and winter, and the diapause time is as long as 180–210 days. Since CLIMEX cannot set both overwinter and oversummer at the same time, we combined the growth index and proposed the “growth curve fitting method” to set diapause parameters. The diapause parameters were adjusted as follows: DPD was set to 120 in combination with the template data. In the study of Wang and Yan [39,40], the light periods are 15 and 13, and we set DPD0 to 14 and DPT0 and DPT1 to 25 and −5, respectively. Due to the diapause to overwinter and oversummer, DPSW was set to 0.

### 2.5. Classification of EI Values

EI (ecoclimatic index) values are commonly categorized to identify the climatic favorability levels for *A. cinerarius* in order to explain the favorability of a place for a species in greater detail. Referring to Li [41], according to the actual distribution and the degree of harm, the EI calculated by the CLIMEX software was divided into the unfavorable region, EI = 0; marginal region 0 < EI ≤ 5; favorable region 5 < EI ≤ 15; and very favorable region EI > 15.

## 3. Results

The result of the model fits the known distribution of *Apocheima cinerarius* well, and 95.9% of the distribution points fall in the area that is simulated as a suitable environment (Figure 1). One point is located in an area where the simulated climatic conditions are not suitable. The main reason is that the soil moisture in the climate data of Alaskan area is 0, which is inconsistent with the actual situation, so after superimposing irrigation, all simulated suitable areas correspond with actual areas. In addition to describing the distribution of known *A. cinerarius*, this model shows the potential to spread in other countries, such as the United States and Canada (Figure 2).

### 3.1. The Impact of Irrigation

Regardless of irrigation, the very favorable, favorable, and marginal regions are mainly in the northern temperate zone (30–50° N) and the southern temperate zone (20–60° S), accounting for 23.81% of the total land area, of which the very favorable, favorable, and marginal region accounted for 2.97%, 7.74%, and 13.09% of the global land area, respectively. After imposing simulated irrigation regimes, the potential suitable areas are significantly reduced. Very favorable regions made up 2.29% of the total global land area, favorable region accounted for 6.04%, and the marginal region accounted for 11.73%. The main areas of change involved parts of Asia and North America (Figure 2 and Figure 3). Although there is no record of the distribution in Europe, America, and Australia, the suitable climate may allow the pest’s introduction into these areas.

### 3.2. America

According to the known distribution, this species does not occur in the Americas, but there are highly suitable areas and a large number of hosts for *A. cinerarius* in the Americas. The marginal regions are mainly concentrated in most of Mexico, the eastern part of Brazil, the northern part of Argentina, and the western part of Peru; the favorable regions are mainly distributed in the western United States and southern Canada, the central part of Argentina, and the transition area between Bolivia and Peru; and the very favorable regions are mostly distributed in the mid-western region of the United States and the central and southern regions of Argentina, and there are also a few in Chile and central Bolivia.

With the impact of climate change, the suitability of *A. cinerarius* expands northward (Figure 4). In the United States, current suitable areas are mainly distributed in the west; with projected climate change low and moderately suitable areas in the west gradually become highly suitable areas (Figure 5). Suitable areas in Canada increase significantly, mainly with expansion to the north. The suitable area in South America is expected to gradually decrease with climate change. With the exception of Argentina, the favorable and marginal regions in other South American countries slowly turn into unsuitable areas, while the marginal regions in northern Argentina slowly change into unsuitable areas, and the favorable and very favorable regions in the central part change marginal regions. Some favorable and marginal regions in northern Colombia and Venezuela were also observed (Figure 6).

### 3.3. Africa

The total suitable area in Africa is relatively low, the proportion is 8.88%, with the main suitable areas are concentrated in the northernmost and southernmost areas of Africa (Figure 7). With climate change, part of Africa becomes essentially unsuitable for the survival of *A. cinerarius* (Figure 6). The very favorable regions in Morocco, Algeria, and South Africa gradually become favorable and marginal regions, while the favorable and marginal regions in Tanzania, Kenya, and Ethiopia become unsuitable.

### 3.4. Asia

In total, 28.59% of the land area of Asia is suitable for *A. cinerarius* (Figure 7). Marginal regions are mainly distributed in southwestern Russia, northern China, and northern Kazakhstan; favorable regions are mainly distributed in most areas of Kazakhstan, Turkmenistan, Uzbekistan, and central China; and very favorable regions are distributed in central China, northern Pakistan, eastern Uzbekistan, and most of Iran.

In Asia, the total suitable area increases with climate change, and the area of marginal and favorable regions increase while the area of very favorable regions decreases, with an overall trend move to the northeast. Due to climate change, the marginal favorable regions in northern North Korea are expected to become favorable. In Russia, in addition to the increase in the area of low-to-moderate suitable areas in the northern region, new suitable areas will be created in the easternmost part with climate change. The climatically suitable regions will have a significant decline in southern Asia, but in Central Asia, especially China, the suitability will significantly increase.

### 3.5. Australia

The total suitable area in Australia accounts for 48.09% of the total land area (Figure 7). The suitability increased from central Australia to the south and then decreased. With climate change, suitability greatly declines. The marginal and favorable regions in central Australia will gradually become unsuitable areas. Many very favorable regions in southern New Zealand will be transformed into low and moderately suitable areas. In addition, many marginal regions in southern Australia become favorable regions.

### 3.6. Europe

In Europe, 26.96% of the total area is suitable for *A. cinerarius* (Figure 7). The marginal regions are distributed in parts of Ukraine, Belarus, Finland, Sweden and Russia; the favorable regions are located in northern Finland, eastern Poland, and central Spain; and the very favorable regions are located in the central and eastern areas of Spain. With climate change, the suitability of central Europe declines. The suitability of the northernmost and southernmost regions of Europe first increase, to mid-century, and then decrease with climate change. For example, the suitability of northern Finland will increase by 2030, and many unsuitability regions and the marginal regions will become favorable areas; by 2100, Finland will become a completely unsuitable area.

### 3.7. Bioclimatology

In the CLIMEX model, the stress index has the greatest impact on the simulated range of species, while the growth index has a greater impact on the simulated suitability model within the range, which in turn drives the simulated phenological model.

We generated the annual growth index of Ningxia and Changji (Figure 8a,b), where a population of *A. cinerarius* is established, and where the population dynamics have been studied. According to field surveys, only March, April, and May are suitable for *A. cinerarius* growth in Changji Hui Autonomous Prefecture, while in Ningxia, the species grows in March, with the index rising rapidly and reaching its maximum in April. It gradually decreases to zero from May to July. The increases in temperature and rainfall are the limiting factors for *A. cinerarius*. We predicted the growth index in Nebraska, Alberta, Spain, and South Hankyong Province in North Korea under future climatic conditions (Figure 8c–f). In Nebraska, *A. cinerarius* will emerge two months earlier than in China, and at the same time, it will reach a growth index (GI) peak at the end of March, and due to the influence of the humidity index, the growth index (GI) will drop to 0 in June. The overall trends in Alberta and Nebraska are similar, but the growth will be delayed by one month in Alberta, mainly due to the relatively low temperature, which cannot meet the emergence demand. In Castile and South Hanjing Province, the occurrence will be advanced to December, where it has a strong growth potential. It can be seen that it will occur in the cooler and drier months, and gradually disappear in the warmer and wetter months.

### 3.8. Driving Variables Limiting the Potential Distribution

According to the EI formula in CLIMEX, when the GI value is 0 or the SI is greater than 100, *A. cinerarius* will not survive in the area. As a species in arid regions, the model indicated *A. cinerarius* is mainly distributed from 20° N to 70° N and 30° S to 50° S. The EI value is mainly affected by four variables that limit its potential distribution: cold stress (CS), heat stress (HS), dry stress (DS), and wet stress (WS).

The results show that these four variables limit the global suitable area. Under historical and future climatic conditions, cold stress is the main restricting factor. When below 70° N, cold stress will prevent the survival of *A. cinerarius* in most parts of Canada, Russia, and Greenland. On the other hand, with climate change, the impact and degree of heat stress will expand and limit the distribution in northern Africa, Australia, and northern South America. In arid regions, moisture stress is the main factor restricting the global distribution of *A. cinerarius*. Almost all regions of the world have moisture stress for the species. However, with climate change, the degree of moisture stress in some areas decreases, which is the main reason for the expansion of suitable area for *A. cinerarius* in some regions. Dry stress mainly has a restrictive effect in southern Xinjiang, China, and northern Africa (Figure 9).

## 4. Discussion

This study reveals that *A. cinerarius* has a wide range of potential suitable areas in many parts of the world, mainly in the United States, Canada, China, Argentina, and Australia under historical climate conditions (Figure 2). Climate change has both positive and negative effects for *A. cinerarius*, and projections indicate that suitability of *A. cinerarius* will increase in the United States and China, and gradually decrease in Australia and Argentina (Figure 7). Cold stress and dry stress limit the distribution in cold regions (e.g., Canada and Russia) and arid regions (e.g., North Africa) of *A. cinerarius* (Figure 9).

### 4.1. Impacts of Climate Change

Climate change can affect the occurrence of pests in forests by altering the structure of forest ecosystems and increasing their vulnerability and susceptibility [42]. Climate change has an impact on pests both directly and indirectly. Changes in the quantity of natural enemies, mutualists, and competitors, as well as direct impacts on pest insect growth and survival, adaption capabilities, host plant availability, and physiological changes in host defenses, might modify pest insect disturbance patterns [43]. Increases in temperature and variations in moisture availability may have direct effects on insect survival and development rates [44], and geographic distributions are anticipated to shift as a result of these changes [45], as an example under the influence of climate change, the latitude and longitude distribution of *Orthotomicus Erosus* (Woll) will change, and it will have a higher survival rate and a faster development rate [46]. Our findings demonstrate that with climate change, rising temperatures will make many currently suitable areas for *A. cinerarius* unsuitable in the future. This is because rising temperatures make the colder northern regions milder. In contrast, we found that with changes in climate scenarios and rising temperatures will make many currently suitable areas unsuitable in the future [47]. The net effect of climate change is to increase the area suitable for *A. cinerarius* worldwide (Figure 4).

### 4.2. Effects of Host Distribution

Climate change causes not only an increase in temperature but also elevated CO_2_ levels, more severe droughts, and more frequent storms. Forest pests, natural enemies, and their hosts will be directly or indirectly affected by climate change [48]. Insect infestations make trees more susceptible to water stress. When insects attack trees, the crown and/or roots are harmed, affecting their ability to regulate water. As a result, trees are more vulnerable to subsequent instances of intense drought, resulting in increased death rates [49].

Climate model uncertainty, as well as natural variability in the climate system, can lead to uncertainty in climate forecasts, impacting prediction outcomes. In addition to temperature change, other factors such as geography, soil type, host plant availability, land use, and human activity may limit pest dispersal [50]. Insects feed on the nutrients that their host plants supply. As a result, the availability of host plants constrains insect distributions. The distribution range of *A. cinerarius* is extremely wide, and one of the reasons for this is that there are many host species. *Apocheima cinerarius* damages as many as 14 natural hosts [51]. Among them, *Sophora japonica* and *Ulmus pumila* are often used in the construction of protection forest in arid environments, which have huge economic and ecological benefits; *P. euphratica* grows on saline–alkali soils to protect oasis and maintain the stability of the oasis ecosystem [6].

The current distribution range of *Sophora japonica*, *P. euphratica*, and *Ulmus pumila*, which are often harmed by *A. cinerarius*, overlaps with the predicted suitable areas under future climatic conditions (Figure 10). For example, we predicted the growth index (GI) for the United States, Spain, North Korea, and northern Canada. This also shows that if it is introduced into other regions, the presence of a host may cause serious harm to these regions and will be a serious challenge for the local ecosystem. On the contrary, some areas are not suitable for the survival of *A. cinerarius* ‘s host, which will limit the spread of *A. cinerarius*.

### 4.3. Process of Diapause

*Apocheima cinerarius* is relatively unique, because it survives both the winter and summer through diapause, and this may also help to explain its potential to be widely distributed. The CLIMEX model can select species diapause parameters, but, unfortunately, the model can only choose winter diapause or summer diapause [26]. Through communication with software developers and after many adjustments, we finally adjusted the parameters based on the growth index and actual life history. Winter diapause and summer diapause are well represented in models (Figure 8), the monthly growth index is in full compliance with its biological characteristics, thereby increasing the accuracy and feasibility of model prediction, and it also provides a feasible solution to this unusual situation in the future. As a species with a long diapause time, *A. cinerarius* can withstand heat stress in the summer and cold stress in the winter. This is also one of the reasons why *A. cinerarius* has spread in China in recent years and has become increasingly harmful.

### 4.4. Limitations of the CLIMEX Model

Wang used MAXENT to analyze and predict the potential distribution areas in Xinjiang’s characteristic fruit forest planting areas [22]. Compared with the results of this study, it was found that the suitable areas for the prediction in the south of Tianshan are similar. However, Wang concluded that the distribution of areas north of Tianshan (such as Changji area) is very small; only the areas of Turpan and Hami have a small number of distributions. This is slightly different from the actual distribution area. Li [52] explored the occurrence in Kuitun City and found that low temperature is the biggest factor affecting *A. cinerarius* emergence by comparing the data over the years, and the occurrence is gradually increasing due to the trend of climate change. It can be seen from Figure 9 that due to climate change, the cold stress north of the Tianshan Mountains in Xinjiang has gradually disappeared, indicating that *A. cinerarius* is increasingly suitable for survival. We believe that differences in models, data sources, and analysis methods may be the main reasons for the differences in results. The MAXENT model needs to use the actual distribution data of species to make predictions [53]. The data source is the data of the orchard planting area. There are many human interference factors, such as watering and the use of pesticides. In order to reduce the difference as much as possible, we superimposed irrigation scenarios. From the prediction results, overlay irrigation will reduce the suitability in Xinjiang but will expand the suitable area in Xinjiang, China. After overlay irrigation, the suitable area includes the Tianshan Mountains. The prediction results for the area south of the mountain range are basically the same as those of Wang, but suitable area to the north of the Tianshan mountain range is also a result. Although we considered the issue of irrigation, there are still some limitations, because the world’s irrigation area data do not include the specific data of each region for each month, and the relationship between pests and multiple hosts is also difficult to fit in the model. These issues may affect the distribution of research results and predictions.

Due to *A. cinerarius* being mainly distributed in China, with limited distribution in Central Asia, the parameters of CLIMEX may not be representative of the parameters of various regions of the world, resulting in differences in prediction results [54].

The CLIMEX model’s predictions are only based on climatic factors, but the relationships between host and natural enemies, topography, land use, and human activities will all have an impact on species distribution, growth, and survival [55]. At the same time, the biological characteristics of species potentially undergo evolutionary changes over time, and these changes may affect forecast distribution. Our study, as is the case for most such climate driven projections, does not take into account the potential for pest evolution. Climate is the only easy-to-handle factor to obtain preliminary and conservative estimates of the potential distribution of species [50]. Recently, the Bayesian SDM using Gaussian process (GP), developed by Golding and Purse [56], has allowed for the user to utilize a prior estimate of a model function to include past ecological information [57]. With the progress of species distribution models, it is recommended to consider additional non-climatic factors in the prediction of future assessments and incorporate other SDMS predictions to better manage invasion risks and enhance the accuracy of predictions [58].

### 4.5. Strategies and Recommendations

Climate change, increased global commerce, and land use change all have the potential to modify insect pest–disturbance relationships. Insect pest outbreaks are likely to become more common as global warming accelerates. Climate change has a significant impact on insect fitness and dispersion [59].

Our research offers theoretical risk assessment and early warning of *A. cinerarius*. Our results indicate that favorable climatic regions occur not only in Asia but also in North America, South America, Australia, Europe, and Africa (Figure 3 and Figure 4). Awareness of this potential can justify improved quarantine and defense measures to avoid the spread of dangerous pests through international trade. Climate adaptability is rising, especially in most parts of North America; as a result, large-scale pest outbreaks are possible in the future, and management and preventative efforts should be enhanced. Furthermore, the results of the simulations under future climate conditions reveal that as a result of climate change, the prospective distribution may vary (Figure 4), with the range of eligible areas increasing to higher latitudes, notably in the northern hemisphere.

## 5. Conclusions

We use known distribution data and meteorological data (mainly temperature, humidity and precipitation, and overlapped irrigation) in China and Central Asia to predict the current and potential suitable areas of *Apocheima cineraria* invasion in the future. The results show that the spread of *A. cinerarius* will move northward, but there are exceptions—the suitability of some northern regions is projected to decline. We also found a negative correlation between the impact of irrigation treatment and the suitable area, which implies that irrigation treatment before the pupal stage may be helpful for pest control. According to the analysis of driving variables, we indicated that cold stress is the main limiting factor under historical and future climate conditions. We also discussed the degree of overlap between the host plant and the potential suitable area; the widely distributed host plants provide favorable conditions for the spread. The example in this article provides a solution for the CLIMEX parameter settings when a species undergoes winter diapause and summer diapause. Finally, our research provides a theoretical basis for preventing the spread of *A. cinerarius* and early monitoring and warning.

## Figures and Tables

**Figure 1 insects-13-00059-f001:**
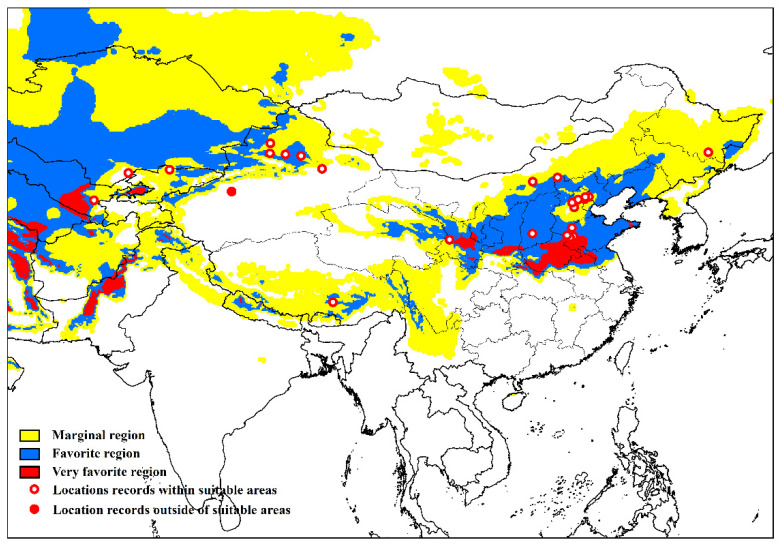
*Apocheima cinerarius* distribution points used to build the model.

**Figure 2 insects-13-00059-f002:**
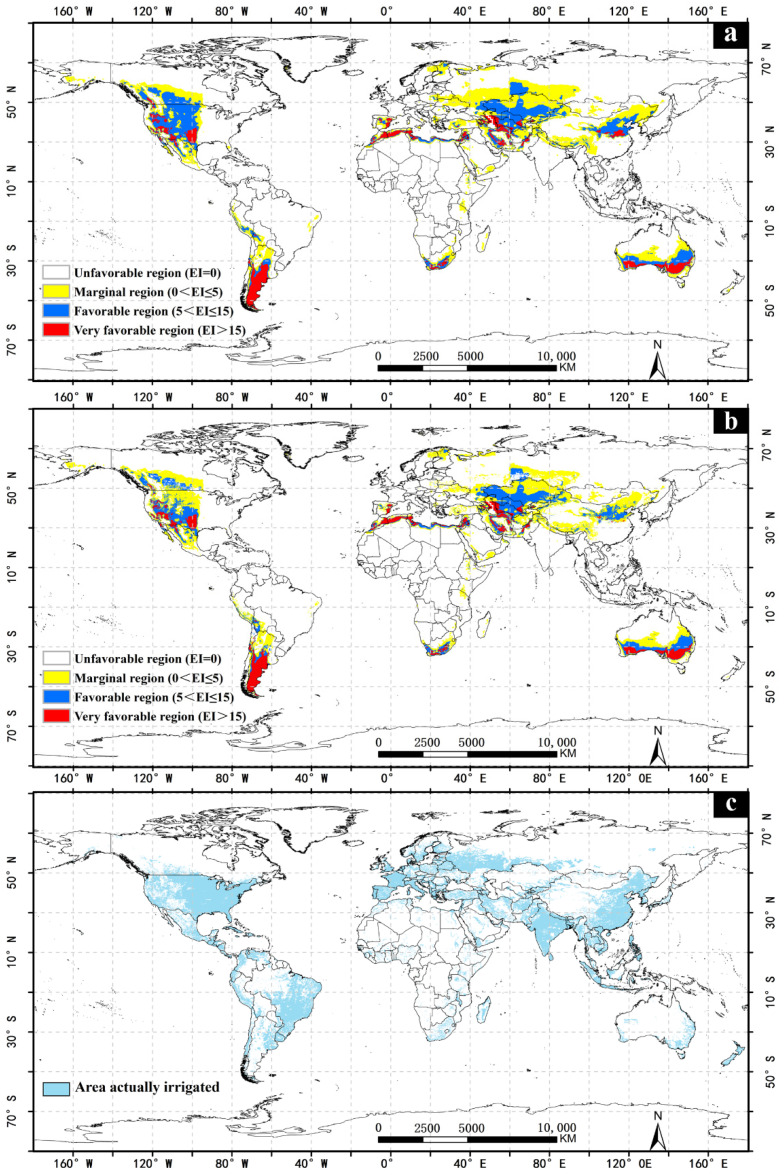
Potential global distribution of *Apocheima cinerarius* under the historical climate. (**a**) Projected global distribution of *Apocheima cinerarius* assuming natural rainfall; (**b**) projected global distribution of *Apocheima cinerarius* with irrigation; (**c**) global map of irrigation areas.

**Figure 3 insects-13-00059-f003:**
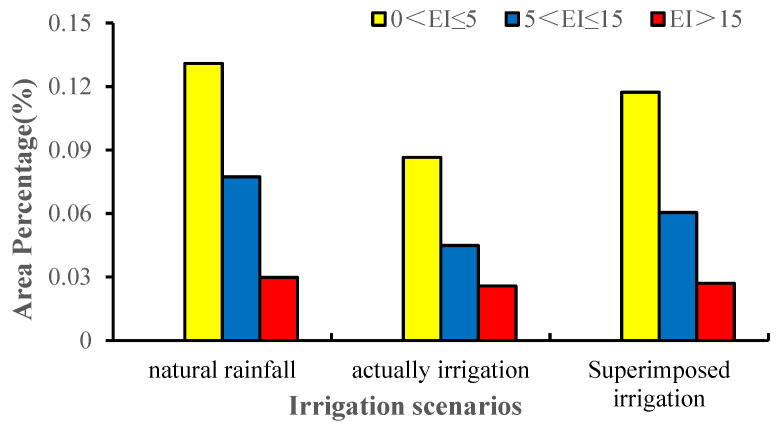
Changes in the potential distribution under historical climate conditions for two types of irrigation.

**Figure 4 insects-13-00059-f004:**
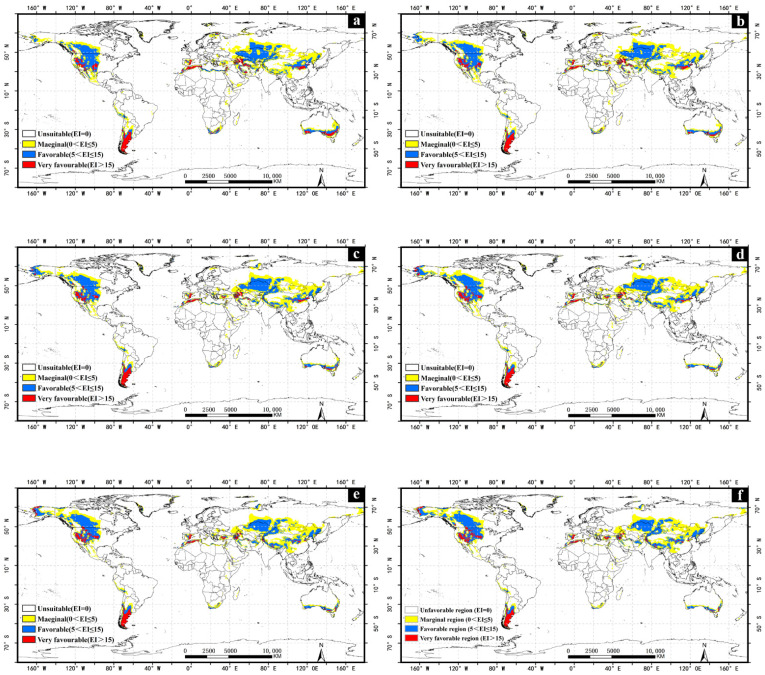
Potential global distribution of *Apocheima cinerarius* under future climate. (**a**) 2030; (**b**) 2050; (**c**) 2070; (**d**) 2080; (**e**) 2090; (**f**) 2100.

**Figure 5 insects-13-00059-f005:**
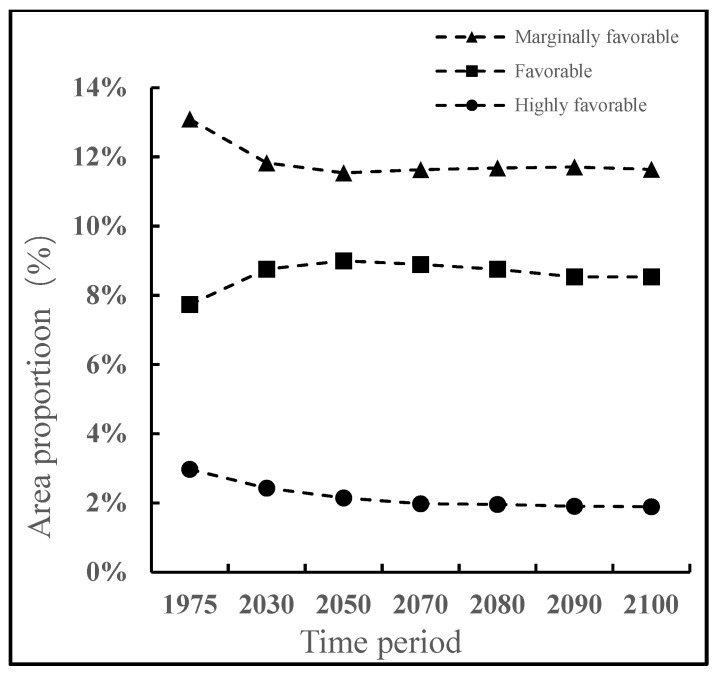
Proportion of the total distribution area of *Apocheima cinerarius*, represented by the various region during different periods.

**Figure 6 insects-13-00059-f006:**
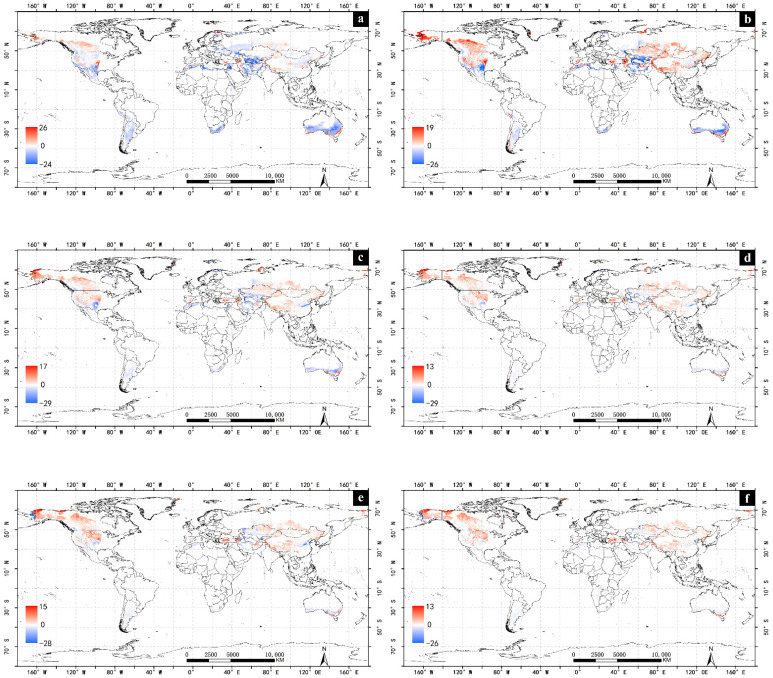
EI difference map of *Apocheima cinerarius* in different periods in the world: (**a**) 2030–1975; (**b**) 2050–2030; (**c**) 2070–2050; (**d**) 2080–2070; (**e**) 2090–2080; (**f**) 2100–2090.

**Figure 7 insects-13-00059-f007:**
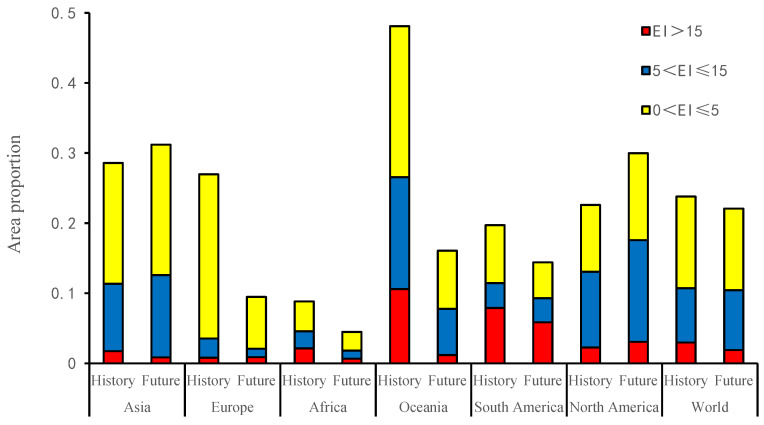
Percentage of the suitable area of *Apocheima cinerarius* in different continents under historical and future conditions.

**Figure 8 insects-13-00059-f008:**
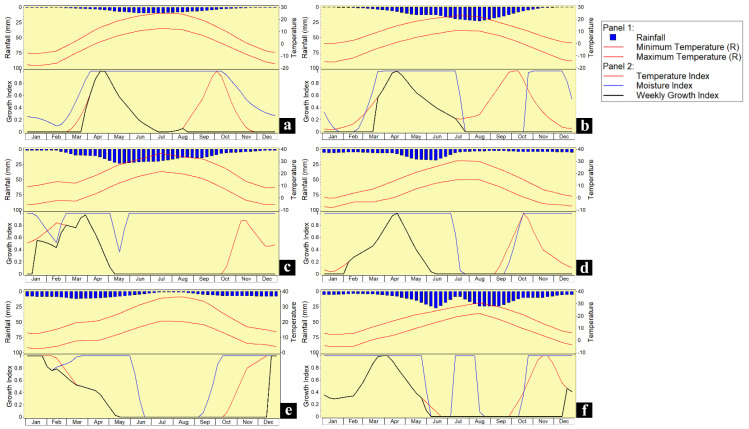
Predicted growth index of *Apocheima cinerarius**:* (**a**) Hutubi; (**b**) Linxia; (**c**) Nebraska; (**d**) Alberta; (**e**) Castilla; (**f**) Nanjing Road.

**Figure 9 insects-13-00059-f009:**
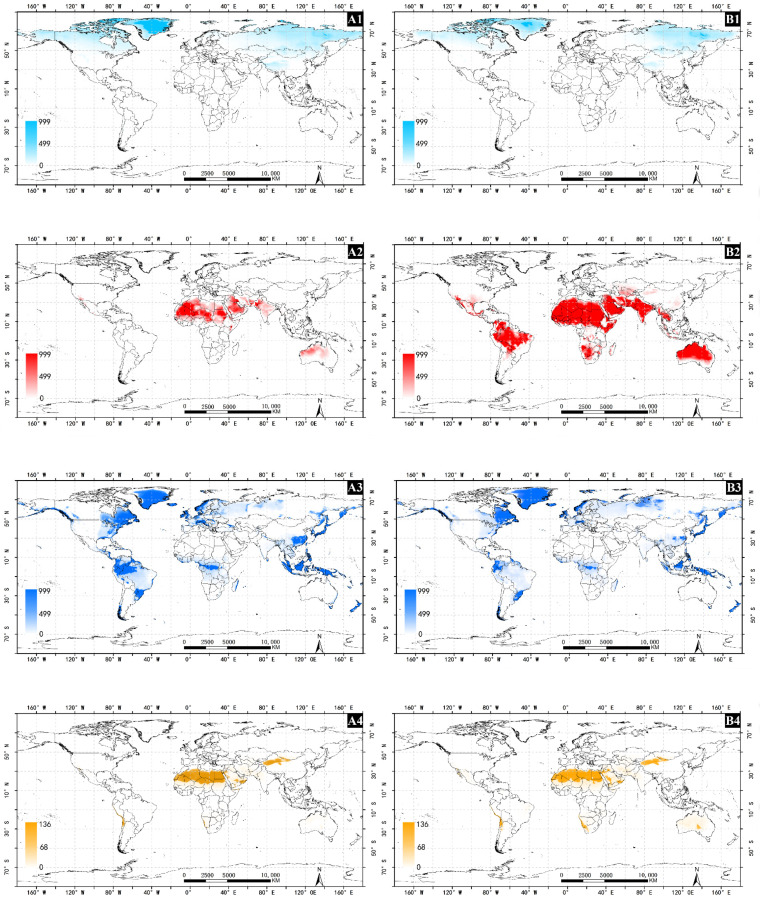
Limiting distribution maps for four different conditions. (**1**), (**2**), (**3**), and (**4**) represent cold stress (CS), heat stress (HS), dry stress (DS) and wet stress (WS), respectively. (**A**) historical climatic conditions; (**B**) future climatic conditions.

**Figure 10 insects-13-00059-f010:**
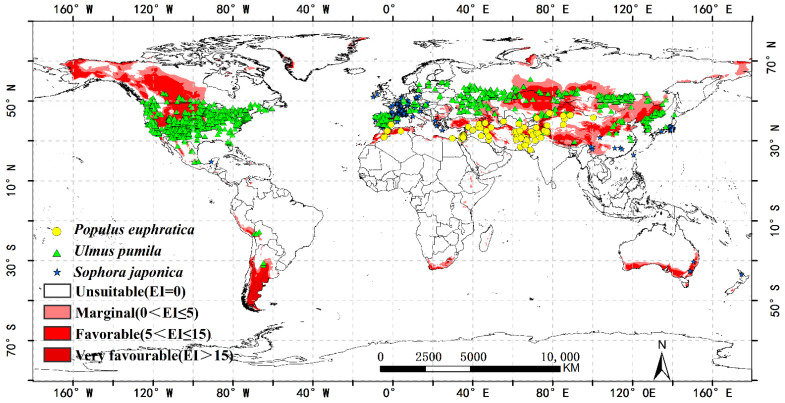
The global potential suitable area of *Apocheima cinerarius* and its main host. The yellow circle, green triangle, and blue star represent *P. euphratica*, *Ulmus pumila*, and *Sophora japonica*, respectively.

**Table 1 insects-13-00059-t001:** CLIMEX parameters of *Apocheima cinerarius*.

CLIMEX Parameter	Description	Value	Unit
DV0	Lower temperature threshold	−2	°C
DV1	Lower optimum temperature	7	°C
DV2	Upper optimum temperature	15	°C
DV3	Upper temperature threshold	25	°C
PDD	Effective accumulated temperature	650	DD
TTCS	Cold stress temperature threshold	−5	°C
THCS	Cold stress temperature rate	0.0002	week^−1^
TTHS	Heat stress temperature threshold	35	°C
THHS	Heat stress temperature rate	0.005	week^−1^
SM0	Lower soil moisture threshold	0.05	*
SM1	Lower optimal soil moisture	0.1	*
SM2	Upper optimal soil moisture	0.4	*
SM3	Upper soil moisture threshold	0.45	*
SMDS	Dry stress threshold	0.05	°C
HDS	Dry stress rate	0.005	week^−1^
SMWS	Wet stress threshold	0.45	°C
HWS	Wet stress rate	0.003	week^−1^
DPD0	Diapause induction day length	14	*
DPT0	Diapause induction temperature	25	°C
DPT1	Diapause termination temperature	−5	°C
DPD	Diapause development days	120	day
DPSW	Diapause summer or winter indicator	0	*

* indicates that there is no unit name.

## Data Availability

Not applicable.

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
