# Peer review of "Climate Change Impacts on the Potential Distribution of Apocheima cinerarius (Erschoff) (Lepidoptera: Geometridae)"

_insects, 2022, doi:10.3390/insects13010059_

Round 1

Reviewer 1 Report

General remarks

The manuscript is an interesting study about the impact of climate change on a pest species in China. It is well written and the conclusion should be interesting for the wider international readership, especially the suitability of future climate scenarios. The results are partly simplified and therefore I suggest to present data for China on one hand which has higher accuracy I suppose, and on the other hand the worldwide situation (not for every continent as shown- because I wouldn’t give strong conclusions about other regions). I would also suggest to give in the Discussion or Conclusion a statement about more specific studies in other regions/parts of the world in the future in order to get higher accuracy and more precise CLIMAX predictions.

Because the result is a prediction of northward movement in the future, I would suggest to give citation about already proofed data and published paper (see below).

Specific remarks

Page 1

Line 42 A. cinerarius should be Italic

Line 44 A. cinerarius

Page 2

Lines 1-12 This should be shortened-there are no reason to write about one specific tree species (P. euphratica) among 14 species attacked by the pest. I understand that it was used by CLIMAX but than how could it be representative globally?

Lines 17-24 because the study is research very often themed you should emphasis the changes that already happened regarding the climate change impact on pests especial the northward movement. I suggest to cite some Mediterranean pests (like Orthotomicus erosus, Thamatopoea pityocampa) because Mediterranean region is a hotspot of climate change; or the huge American problem Dentroctonus ponderosae. These species moved northwards and that shall vindicate your study.

Page 3

Line 2 I noticed in many sentences that interspace is lacking (see Line 12, Page 4 Line 17, Page 5 Line 1, 10, 15….please be careful about that throughout your text

Page 15

Line 26 Wang-Is that the reference 26? If yes you should give the reference, please rewrite this part

Page 16

Line 3  - time

Author Response

Dear reviewer:

Re:Manuscript ID:insects-1474999

  On behalf of my co-authors, we thank you very much for giving us an opportunity to revise our manuscript. We appreciate editor and reviewers very much for their positive and constructive comments and suggestions on our manuscript entitled “Climate change impacts on the potential distribution of Apocheima cinerarius” (ID: insects-1474999). We have considered the comments very carefully and have revised the paper accordingly.

Response to Reviewer 1

[General Comment] The manuscript is an interesting study about the impact of climate change on a pest species in China. It is well written and the conclusion should be interesting for the wider international readership, especially the suitability of future climate scenarios. The results are partly simplified and therefore I suggest to present data for China on one hand which has higher accuracy I suppose, and on the other hand the worldwide situation (not for every continent as shown- because I wouldn’t give strong conclusions about other regions). I would also suggest to give in the Discussion or Conclusion a statement about more specific studies in other regions/parts of the world in the future in order to get higher accuracy and more precise CLIMAX predictions. Because the result is a prediction of northward movement in the future, I would suggest to give citation about already proofed data and published paper (see below).

Response: Thank you very much!

[Response to comments 1]: (Page 2 Lines 1-12 This should be shortened-there are no reason to write about one specific tree species (P. euphratica) among 14 species attacked by the pest. I understand that it was used by CLIMAX but than how could it be representative globally?)

Response: We thank you for the critical comments and helpful suggestions. A lot of researches are carried out on Populus euphratica and A. cinerarius, such as reference 5, 6, and 7. At the same time, Populus euphratica has important ecological value, A. cinerarius that harms it will have more research significance. From a global perspective, Populus euphratica is not only distributed in China, but also in many Asian regions and northern Africa. The biological characteristics of Populus euphratica also help us when setting CLIMEX parameters, so we think we should introduce Populus euphratica appropriately.

[Response to comments 2]: (Page 15 Line 26 Wang-Is that the reference 26? If yes you should give the reference, please rewrite this part.)

Response: We are very sorry for this problem, it may be an omission in setting the reference at that time, we have corrected this error. This sentence refers to the reference 17.

[Response to comments 3]: (because the study is research very often themed you should emphasis the changes that already happened regarding the climate change impact on pests especial the northward movement. I suggest to cite some Mediterranean pests (like Orthotomicus erosus, Thamatopoea pityocampa) because Mediterranean region is a hotspot of climate change; or the huge American problem Dentroctonus ponderosae. These species moved northwards and that shall vindicate your study.)

Response: This suggestion is very useful, and it also matches our results. We have included relevant literature for discussion in Section 4.1.

[Response to comments 4]: (Page 1 Line 42 A. cinerarius should be Italic; Line 44 A. cinerarius; Page 3 Line 2 I noticed in many sentences that interspace is lacking (see Line 12, Page 4 Line 17, Page 5 Line 1, 10, 15…. please be careful about that throughout your text); Page 16 Line 3 – time.)

Response:This kind of problem shouldn't happen at all! We have carefully checked the similar errors from beginning to end and have all corrected them. Revised portion are marked up using the “Track Changes” function.

Reviewer 2 Report

  1. General Comments

The paper presented has a good structure and is very easy to read and to understand. The model presented is a theorical one, presents distinct scenarios of the Apocheima cinerarius dissemination well supported by studies already done and in climatic parameters. So, for my point of view, it is an excellent exercise allowing the establishment of managements measures.

  1. Section by section

2.1. Introduction:

Introduction is very easy to read, very comprehensible and has a lot of references to consolidate the affirmations made.

2.2. Material and Methods:

Material and Methods are very clear and allow to understand the study. In case of interest or necessity the description of methods used allows to replicate the assay.

2.3. Results:

Results are well presented; graphic component gives reliable information easy to interpretate. 

2.4. Discussion:

Discussion is well conducted and very interesting to read.

  1. Suggestions

Despite the interest of the presented paper, some small mistakes have been detected and some parts of the article could be improved.

3.1. Point by point of small mistakes

  1. a) Page 3 – Line 8.

In is in capital letters however there is not a point before. Please correct the mistake.

  1. b) Page 15 - Line 26.

The reference of Wang paper is missing. Please correct the mistake.

3.2. Point by point of improvement suggestions

  1. a) In the text the description of A. cinerarius life cycle could be improved. It will be interesting to provide more information about pupae development to understand the influence of soil moisture.

  1. b) In Discussion, page 15, I don´t like to read the conversation with software developers. I suggest remove these sentences, starting in line 10 (by communicating) until line 14 (Therefore).

Author Response

Dear reviewer:

Re:Manuscript ID:insects-1474999

  On behalf of my co-authors, we thank you very much for giving us an opportunity to revise our manuscript. We appreciate editor and reviewers very much for their positive and constructive comments and suggestions on our manuscript entitled “Climate change impacts on the potential distribution of Apocheima cinerarius” (ID: insects-1474999). We have considered the comments very carefully and have revised the paper accordingly.

Response to Reviewer 2

[General Comment] The paper presented has a good structure and is very easy to read and to understand. The model presented is a theorical one, presents distinct scenarios of the Apocheima cinerarius dissemination well supported by studies already done and in climatic parameters. So, for my point of view, it is an excellent exercise allowing the establishment of managements measures.

Response: Thank you very much!

[Response to comments 1]: (b. In Discussion, page 15, I don´t like to read the conversation with software developers. I suggest remove these sentences, starting in line 10 (by communicating) until line 14 (Therefore).)

Response: We thank you for the critical comments and helpful suggestions. We have carefully considered this suggestion, and readers seem to be not interested in the details. We modified the expression to show the results directly to the readers.

[Response to comments 2]: (a. Page 3 – Line 8. In is in capital letters however there is not a point before. Please correct the mistake. b. Page 15 - Line 26. The reference of Wang paper is missing. Please correct the mistake.)

Response: We are very sorry for this problem, it may be an omission in setting the reference at that time, we have corrected this error. This sentence refers to the reference 17.

[Response to comments 3]: (a. In the text the description of A. cinerarius life cycle could be improved. It will be interesting to provide more information about pupae development to understand the influence of soil moisture.)

Response: Thank you for your good suggestion. We have improved the expression of life history to make it more complete and easy to understand. Regarding the relationship between pupae and soil moisture, we have carried out a more detailed introduction in section 2.4.2, and how to use this information for parameter settings.

Reviewer 3 Report

I found the manuscript "Climate change impacts on the potential distribution of Apocheima cinerarias" interesting and potentially very good.

Unfortunately there are several typewriting mistakes and the English must be reviewed. 

Moreover, in the introduction, I am suggesting to provide more information; in particular, there is a paper (Liu et al., 2015: Evolutionary history of Apocheima cinerarias, a female moth in northern China) that is describing two different genotypes, with different distribution in China. Considering that the species has flightless females, the situation should take under consideration other aspects then only climatic issues. 

The manuscript needs an important English revision. Moreover, there are nomenclature mistakes. At the beginning of my revision, I started to make corrections on the manuscript, but at the end I decided that most of the mistakes are repeated so many times: so I stopped to include the same correction in the full document (but the authors should do it). 

Discussion and  Conclusion chapters are too small and superficial: definitely they must be improved.

Author Response

Dear reviewer:

Re:Manuscript ID:insects-1474999

  On behalf of my co-authors, we thank you very much for giving us an opportunity to revise our manuscript. We appreciate editor and reviewers very much for their positive and constructive comments and suggestions on our manuscript entitled “Climate change impacts on the potential distribution of Apocheima cinerarius” (ID: insects-1474999). We have considered the comments very carefully and have revised the paper accordingly.

Response to Reviewer 3

[General Comment] I found the manuscript "Climate change impacts on the potential distribution of Apocheima cinerarias" interesting and potentially very good.

Response: Thank you very much!

[Response to comments 1]: (in the introduction, I am suggesting to provide more information; in particular, there is a paper (Liu et al., 2015: Evolutionary history of Apocheima cinerarias, a female moth in northern China) that is describing two different genotypes, with different distribution in China. Considering that the species has flightless females, the situation should take under consideration other aspects then only climatic issues.)

Response: We thank you for the critical comments and helpful suggestions. The document helps us explain the spread of Apocheima cinerarias, and we have cited this document in the introduction.

[Response to comments 2]: (The manuscript needs an important English revision. Moreover, there are nomenclature mistakes. At the beginning of my revision, I started to make corrections on the manuscript, but at the end I decided that most of the mistakes are repeated so many times: so I stopped to include the same correction in the full document (but the authors should do it).)

Response: Lastly, we revised the whole manuscript carefully to avoid language errors. In addition, we consulted a professional editing service and asked several colleagues who are native English speakers to check the English. We believe that the language is now acceptable for the review process.

[Response to comments 3]: (Discussion and Conclusion chapters are too small and superficial: definitely they must be improved.)

Response: In Section 4.1, we added theories and empirical studies on the effects of climate factors on insects. At the same time, the previous conclusion may not be well written. We reorganized the information in the article and rewritten a more specific conclusion.

Reviewer 4 Report

This paper reports about the impact of climate change on the distribution of lepidopteran pest Apochemia cinerarius in different continents. The paper is well structured, the aim of the paper is well described and I have no comment on English. The authors used suitable methods (softwares) for obtaining the results, which they presented in proper manner. I suggest to publish a paper after minor revision, since I suggest to correct the paper in the following parts:

Title

I suggest to add the common name and taxonomy of the pest as follows: "Climate change impacts on the potential distribution of poplar looper (Apochemia cinerarius [Erschoff], Lepidoptera, Geometridae)"

Abstract and other parts of the paper

"Erschoff" should be written within parentheses!
In the scienfific literature two names are used for the species in question, namely Apochemia cinerarius and Apochemia cinerarium. CABI for example prefer Apochemia cinerarium! If the authors will use their first solution, then they need to explain this in the text!

Introduction
p.1, line 42: "A. cineraius" should be written in italic!
p. 2, line 1: After "Populus euphratica" the author of the species should be written, so the suitable Latin name is "Populus euphratica Oliv.". This should be taken into consideration for all the species, when they are first mentioned in the text!
p. 2, lines 15-19: I strongly suggest to add and briefly discuss some references about the impact of climate change on distribution on insect. For example:  BERGANT et al., 2005. Impact of climate change on developmental dynamics of Thrips tabaci (Thysanoptera: Thripidae): can it be quantified?. Environmental entomology, 34, 4: 755-766.  

Discussion
p. 15, line 7: replace "he" with "it"

Author Response

Dear reviewer:

Re:Manuscript ID:insects-1474999

  On behalf of my co-authors, we thank you very much for giving us an opportunity to revise our manuscript. We appreciate editor and reviewers very much for their positive and constructive comments and suggestions on our manuscript entitled “Climate change impacts on the potential distribution of Apocheima cinerarius” (ID: insects-1474999). We have considered the comments very carefully and have revised the paper accordingly.

Response to Reviewer 4

[General Comment] This paper reports about the impact of climate change on the distribution of lepidopteran pest Apochemia cinerarius in different continents. The paper is well structured, the aim of the paper is well described and I have no comment on English. The authors used suitable methods (softwares) for obtaining the results, which they presented in proper manner. I suggest to publish a paper after minor revision.

Response: Thank you very much!

[Response to comments 1]: (Title: I suggest to add the common name and taxonomy of the pest as follows: "Climate change impacts on the potential distribution of poplar looper (Apochemia cinerarius [Erschoff], Lepidoptera, Geometridae)")

Response: We thank you for the critical comments and helpful suggestions. The title of the article is really difficult to determine. Your suggestion is great. Adding taxonomy of the pest can help more readers find the content they need more quickly and accurately. We changed the title of the article as follows: “Climate change impacts on the potential distribution of Apochemia cinerarius (Lepidoptera: Geometridae)”. The reason why the common name is not used is that this article is only mentioned in the introduction. At the same time, we also believe that people who need this article can find it by Latin name or taxonomy of the pest.

[Response to comments 2]: (Abstract and other parts of the paper "Erschoff" should be written within parentheses! In the scienfific literature two names are used for the species in question, namely Apochemia cinerarius and Apochemia cinerarium. CABI for example prefer Apochemia cinerarium! If the authors will use their first solution, then they need to explain this in the text!)

Response: Thanks your professional suggestions. Previously we described the species name based on Ning. Later, we checked some papers and found that this is indeed should be written within parentheses. We are grateful for your comments!

Reference: Ning H, Tang M, Chen H. Mapping Invasion Potential of the Pest from Central Asia, Trypophloeus klimeschi (Coleoptera: Curculionidae: Scolytinae), in the Shelter Forests of Northwest China [J]. Insects, 2021, 12(3): 242.

[Response to comments 3]: (p. 2, lines 15-19: I strongly suggest to add and briefly discuss some references about the impact of climate change on distribution on insect. For example:  BERGANT et al., 2005. Impact of climate change on developmental dynamics of Thrips tabaci (Thysanoptera: Thripidae): can it be quantified? Environmental entomology, 34, 4: 755-766.)

Response: We have carried out more literature reading and enriched the content of the “impact of climate change on insects” in the introduction. At the same time, we discussed the impact of climate change in more depth in Section 4.1. The modified place has been marked in red.

[Response to comments 4]: (Introduction: p.1, line 42: "A. cineraius" should be written in italic! p. 2, line 1: After "Populus euphratica" the author of the species should be written, so the suitable Latin name is "Populus euphratica Oliv.". This should be taken into consideration for all the species, when they are first mentioned in the text! p. 15, line 7: replace "he" with "it")

Response: We are very sorry for this problem, it may be an omission in setting the reference at that time, we have corrected this error. Revised portion are marked in red in the paper and marked up using the “Track Changes” function.

Round 2

Reviewer 1 Report

The authors improved their paper significantly. I appreciate citations which confirm the main idea of the manuscript like Ayares et al. 2021 (ref. 40). With reference 41 and 42 it gives a very good background. I suggest only to cite also a specific and current case (e.g. changes in bionomy in recent Outbreaks of Orthotomicus erosus in Mediterranean region; I think this would be very useful giving a very specific example).

Author Response

Dear reviewer:
  Thank you for your affirmation of our work. We are glad to have your further valuable comments on the manuscript: it does lack a concrete example to prove our point. We have added relevant examples based on your suggestions.

Reference: Pernek M, Lacković N, Lukić I, et al. Outbreak of Orthotomicus erosus (Coleoptera, Curculionidae) on Aleppo pine in the Mediterranean region in Croatia[J]. South-east European forestry: SEEFOR, 2019, 10(1): 19-27.

Best wishes,

Weicheng Ding